# Validation and Psychometric Properties of the Polish Version of the Skin Picking Scale-Revised

**DOI:** 10.3390/ijerph19052578

**Published:** 2022-02-23

**Authors:** Joanna Kłosowska, Katarzyna Prochwicz, Dominika Sznajder, Rachela Antosz-Rekucka, Aleksandra Tuleja, Aleksandra Dembińska, Ivar Snorrason

**Affiliations:** 1Institute of Psychology, Jagiellonian University, ul. Ingardena 6, 30-060 Kraków, Poland; joanna.klosowska@uj.edu.pl (J.K.); r.antosz-rekucka@doctoral.uj.edu.pl (R.A.-R.); a.tuleja@student.uj.edu.pl (A.T.); 2Institute of Psychology, Pedagogical University, ul. Podchorążych 2, 30-084 Kraków, Poland; dominika.sznajder@up.krakow.pl; 3Krzysztof Czuma Psychiatry Centre, ul. Korczaka 27, 40-340 Katowice, Poland; aleksandra@dembinska.pl; 4Center for OCD & Related Disorders (CORD), Massachusetts General Hospital, 185 Cambridge Street, Boston, MA 02114, USA; ivarsnorrason@gmail.com; 5Department of Psychiatry, Harvard Medical School, Boston, MA 02115, USA

**Keywords:** Skin Picking Scale-Revised, skin picking, validity, reliability, factor structure, diagnostic accuracy

## Abstract

The Skin Picking Scale-Revised (SPS-R) is an 8-item self-report measure of skin picking behaviors. It includes two subscales related to skin picking symptom severity and picking-related impairments. The study aimed to assess the psychometric properties of the Polish version of the SPS-R in a sample of adults reporting skin picking. The sample of 764 participants was recruited from the general population through an online survey. Among them, 159 meet the criteria of pathological skin picking applied in the original SPS-R validation study, and 57 endorsed all of the DSM-5 criteria for excoriation disorder. The SPS-R was back-translated into Polish. Factor structure, reliability, convergent and divergent validity, and diagnostic accuracy were assessed. Confirmatory factor analyses revealed a two-factor structure of the scale. High internal consistency and convergent and divergent validity were confirmed for the total score as well as for the subscales. High prognostic ability of the SPS-R total score was also demonstrated using ROC analysis: ≥5 was accepted as an optimal cut-off point for distinguishing skin picking sufferers from healthy controls. The Polish version of the SPS-R shows good psychometric properties and appears to be a reliable measure of skin picking symptoms and picking-related impairment.

## 1. Introduction

Pathological skin-picking is a chronic condition whose prevalence in the community is estimated to be surprisingly high. Severe, clinically relevant skin picking affects about 1–5% of the general population [1,2], however, its subclinical forms are reported in 60% of adults [1] and up to 90% of university students [3,4] with female predominance across study samples [5,6].

Skin picking has recently gained more attention due to the introduction of explicit diagnostic criteria of skin picking disorder (SPD) in the Fifth Edition of Diagnostic and Statistical Manual of Mental Disorders (DSM-5) [7]. According to DSM-5, skin picking (excoriation) disorder is diagnosed by the presence of the following features: recurrent skin picking resulting in skin lesions; repeated attempts to decrease or stop picking; clinically significant distress or impairment in social, occupational, or other important areas of functioning caused by picking; absence of another mental or medical condition which may better explain the symptoms.

Apart from medical examination and clinical interview, the presence of SPD can be assessed on clinician-administered pen-paper rating scales, such as Yale–Brown Obsessive-Compulsive Scale modified for neurotic excoriation [8] and the Skin Picking Treatment Scale [9]. Moreover, some brief self-report scales to measure subjective ratings of skin picking behaviors have been developed and are successfully applied in scientific research. Among these, the most widely used is the six-item Skin Picking Scale (SPS; [10]) and its subsequent revision—the eight-item Skin Picking Scale-Revised (SPS-R; [11]). The revised version is a particularly beneficial diagnostic tool because it allows to measure the SPD symptom severity and picking-related impairment separately in two independent dimensions. Hence, it offers the opportunity to examine factors influencing skin picking consequences independently from the disorder course and severity [11]. Moreover, the SPS-R covers all the impairment domains listed in the current SPD diagnostic criteria, i.e., skin damage, distress, and functional impairment [11], and provides a more complete clinical picture of skin picking disorder.

The English version of the Skin Picking Scale-Revised [11] has acceptable psychometric properties, with Cronbach’s alpha coefficients of 0.83 for the total score, 0.81 for the severity subscale, and 0.79 for the impairment subscale. The impairment subscale also reveals significant associations with validated scales measuring disability (Sheehan Disability Scale; SDS [12]) and psychopathology (Depression, Anxiety Stress Scales 21-item version; DASS-21, [13]) which was demonstrated by correlation analyses [11].

In the current study, we aimed to develop a Polish version of the Skin Picking Scale-Revised (SPD-R) and evaluate its psychometric properties. We administered the translation of the SPS-R to a large sample of adults recruited from the general population. Based on the obtained data, we examined internal consistency, factor structure, convergent and divergent validity, and diagnostic accuracy of the scale. Given the findings from the original validation study [11], we expected good psychometric properties and two-factor structure of the tool.

## 2. Materials and Methods

### 2.1. Participants

Initially, 764 participants completed the survey. All of them identified themselves as white. A total of 159 of them (88.7% women, 9.4% men, 1.9% not reported) aged 18–54 (M = 23.58, SD = 6.29) endorsed meeting the criteria for pathological skin picking applied in the original SPS-R validation study [11]: (1) current skin picking resulting in tissue damage; (2) at least mild impairment due to picking; (3) skin picking behavior is not accounted for by the use of drugs, dermatological disease, or psychiatric diseases. The data obtained from this subsample were used to examine the descriptive statistics, factor structure, reliability, and convergent and divergent validity of the SPS-R. The sociodemographic characteristics of this sample are presented in Table 1.

Additionally, to verify the ability of the SPS-R to distinguish individuals with excoriation disorder from healthy individuals an additional subsample of 57 individuals (96.49% women, 3.51% men) aged 18–44 (M = 23.44, SD = 5.27) that endorsed all the DSM-5 criteria for excoriation disorder were derived from the sample of skin picking participants. These individuals were compared to 263 responders derived from the initial sample of 764 participants (89.35% women, 10.65% men), aged 18–65 years (M = 25.45, SD = 9.30), who did not report any dermatological and mental illness or using drugs in the previous six months (see Figure 1). These participants did not differ significantly from individuals meeting the criteria of SPD with respect to age: U(N_healthy_ = 260, N_skin picking disorder_ = 57) = 6793.00, z = −0.99, *p* = 0.32; sex: χ^2^(1) = 2.81, *p* = 0.09; marital status: χ^2^(5) = 3.67, *p* = 0.60; income: χ^2^(4) = 2.77, *p* = 0.60; and education level: χ^2^(8) = 14.84, *p* = 0.06.

### 2.2. Measures

#### 2.2.1. Skin Picking Scale Revised (SPS-R)

The SPS-R [11] is an eight-item self-administered tool evaluating past week skin picking symptoms across two domains: symptom severity domain including four items assessing: (1) the frequency of the urge to pick the skin; (2) the intensity of the urge to pick; (3) time spent on skin picking per day; (4) control over the picking behaviors/the degree one can stop picking; and the functional impairment domain composed of four items related to: (5) emotional distress one experiences from picking; (6) interference with social, work, or role functioning; (7) avoidance of various activities due to picking; and (8) the degree of skin damage as a result of picking.

Participants refer to each item using a five-point rating scale, where the points are assigned a short description reflecting the severity of a particular behavior/experience during one week prior testing (scoring from 0 = no behavior/experience to 4 = extreme behavior/experience). The total score ranges from 0 to 32, with higher scores indicating more severe skin picking.

The Polish translation of the SPS-R was developed for the current study. The forward and back translation were performed. First, the original English version of the scale was independently translated into Polish by two professional translators experienced in psychological translation. These versions were merged during the consensus meeting and were drawn together into one single document with a few corrections for clarity. Afterwards, back-translation into English was performed by psychologists fluent in English and Polish who had not been involved in the original translation. The forward and back translations were then compared to each other and to the original version of the scale. Any inconsistencies were analyzed and discussed. The final Polish version was accepted after small revisions in terms of style. The Polish version fully corresponded to the original scale with respect to instruction, item content, and response style. The online version of the SPS-R was designed alongside the pen-and-paper form.

#### 2.2.2. Depression Anxiety Stress Scales 21-Item Version (DASS-21)

The DASS-21 [13] is a self-report measure which provides scores for assessing the current symptoms of depression (dysphoria, hopelessness, devaluation of life, self-deprecation, lack of interest/involvement, anhedonia, inertia), anxiety (autonomic arousal, skeletal muscle effects, situational anxiety, and subjective experience of anxious affect), and stress (difficulty relaxing, nervous arousal, and being easily upset/agitated, irritable/over-reactive, impatient) on three seven-item subscales. The individual items are rated on a four-point scale ranging from 0 (did not apply to me at all) to 3 (applied to me very much, or most of the time) resulting in the minimum score of 0 and the maximum score of 42 for each dimension. For the study sample (N = 764), the values of Cronbach’s alpha coefficients for depression, anxiety, and stress subscale were 0.92, 0.85, and 0.90, respectively, and 0.95 for the total score. The results of the CFA (with diagonally weighted least squares estimation) conducted in the current sample showed that the three-factor model for the DASS-21 provides a good fit to the data: χ^2^/df = 1.59, CFI = 1.00, NFI = 0.99, RMSEA = 0.03.

#### 2.2.3. Obsessive-Compulsive Inventory-Revised (OCI-R)

The OCI-R [14,15] is an 18-item self-administered measure which assesses the frequency and degree of distress associated with obsessions and compulsions across six three-item subscales measuring separate OCD symptoms: (1) washing, (2) checking/doubting, (3) obsessing, (4) mental neutralizing, (5) ordering, and (6) hoarding. Participants rate the degree to which they were bothered or distressed by the symptoms in the past month on a 5-point scale from 0 (not at all) to 4 (extremely). In the study, we used the Polish version of the OCI-R with Cronbach’s alpha of 0.85 for the total score, and Cronbach’s alphas for the subscales ranging from 0.62 for the mental neutralizing subscale to 0.73 for the washing dimension [15]. CFA (diagonally weighted least squares estimation) ran in the current sample showed that the model with six first-order factors and one second-order factor provides good fit to the data (χ^2^/df = 0.91, CFI = 1.00, NFI = 0.99, RMSEA = 0.00). Internal consistency of the scale in the studied sample was 0.89 for the total score, 0.72 for the washing subscale, 0.90 for the obsessing subscale, 0.72 for the hoarding subscale, 0.83 for the ordering subscale, 0.79 for the checking subscale, and 0.59 for the mental neutralizing subscale. Due to the relatively low internal consistency of the mental neutralizing subscale, and the fact that we were only interested in the general level of obsessive-compulsive symptoms, in the current study, we used only the total score of OCI-R.

#### 2.2.4. Skin Picking Disorder Diagnostic Criteria

In the present study, all the participants were asked questions regarding the presence of the skin picking disorder diagnostic criteria according to DSM-5 [7]. They were asked whether they currently: (1) pick the skin to a degree leading to skin lesions; (2) have made repeated attempts to stop picking; (3) experience significant distress or impairment in social, occupational, or other important areas of functioning due to skin picking; (4) were diagnosed with a dermatological or mental disorder which may underlie their picking behaviors. Questions were answered using a dichotomous yes/no format, except for the question regarding a dermatological illness where the additional ‘probably yes’ and ‘probably no’ responses were included.

#### 2.2.5. Demographic and Illness History Data Sheet

In the study, we also collected data concerning: (1) the demographic characteristics of the participants: gender, age in years, race, marital status, education, employment, monthly income, and place of residence; and (2) psychiatric and dermatological illness history: being formally diagnosed, hospitalized, treated for psychiatric conditions, use of psychoactive substances during six months prior to the study, and being diagnosed with a dermatological illness. Participants responded by choosing one or more answers from the listed response options, except for the question concerning a dermatological illness in case of which participants were required to answer in their own words.

### 2.3. Procedure

Data were collected through an online survey. Potential responders were provided with a link to the study form through online forums and websites that target university samples. As we wanted to capture attention of as many individuals meeting the criteria of skin picking disorder as possible, we included the term “skin picking behaviors (dermatillomania)” in the title of the research as well as in the invitation to participate. To be eligible for the study, participants were required to be 18 years or older and to be fluent in Polish. Current engagement in skin picking behaviors was not necessary to participate in the study: at a later stage, the sample was divided into subsamples of healthy controls and individuals suffering from skin picking, based on their answers to the survey questions (see Figure 1for details). Participation was anonymous and voluntary, no financial remuneration was offered. At the first stage of the survey, participants were provided with short, written information concerning the purpose (“validation of the scale assessing skin picking behaviors”) and length of the research (about 15 min), as well as data processing and privacy. They were also asked to consent to participation by selecting the appropriate box on the computerized study form. Individuals who agreed to participate completed online questionnaires after filling out instructions and answering questions on demographics and history of psychiatric diagnosis. The online system checked whether all questions were answered. Participants could resign from the study at any time without providing a reason. All procedures were reviewed and accepted by the local ethics committee.

### 2.4. Determination of the Sample Size

The minimal sample size required to conduct a confirmatory factor analysis was determined before the study using the A-priori Sample Size Calculator for Structural Equation Models [16]. This software allows to determine both the minimum sample size required to detect the specified effect in the structural equation modeling, and the minimum sample size required in light of the structural complexity of the model. It was determined that, given the eight observed and two latent variables, a minimum of 100 participants suffering from skin picking were needed, taking into account the model structure and ability to detect medium effect size (0.30) with power = 0.80. Bearing in mind the previous study which showed that about 35% of young adults in Poland pick the skin to the extent that it causes visible skin lesions [17], a minimal sample of 300 participants [11] was established.

### 2.5. Analysis Plan

Data were analyzed using JASP (version 0.13.1.0) and JAMOVI (version 1.1.9.0) statistical software. The internal consistency of each subscale was assessed using Cronbach’s α and item-total correlations. The factor structure was evaluated using a confirmatory factor analysis (CFA). Taking into consideration the sample size and ordinal character of the data, the robust diagonally weighted least square (DWLS) estimation method was used. The following fit indices were used to assess the model: chi square, chi square/df ratio, NFI, CFI, and RMSEA. Non-significant chi square test, chi square/df ratio ≤ 2, NFI and CFI values equal or higher than 0.95 and RMSEA value of 0.06 or lower were considered indicative of a good fit of the model [18,19]. Criterion validity was assessed using Spearman’s rank order correlations between the total score of the SPS-R and the score representing number of DSM-5 criteria that were checked by the participants. Divergent validity was assessed using Spearman’s correlations between the SPS-R and the DASS-21 scale and OCI-R scale. The diagnostic accuracy of the SPS-R scale in differentiating healthy individuals from those meeting excoriation disorder’s DSM-5 criteria was assessed via the Receiver Operating Characteristic (ROC) analysis.

## 3. Results

### 3.1. Descriptive Statistics

Descriptive statistics of the measures are presented in Table 2.

### 3.2. Factor Structure

The two-factor structure of the SPS-R, found in the original English version [11], was assessed with the CFA. Fit indices showed excellent fit of the two-factor model (χ^2^ = 17.86, *p* = 0.53; χ^2^/df = 0.940, CFI = 1.00, NFI = 0.97, RMSEA = 0.00). The correlation between latent factors was high (r = 0.71, *p* < 0.001). Factor loadings are presented in Table 3.

Additionally, we assessed the fit of the one-factor model. The chi square test proved to be significant and neither RMSEA nor NFI values fell within the range of acceptable fit. The fit indices were as follows: χ^2^ = 32.75, *p* = 0.036; χ^2^/df = 1.638, CFI = 0.979, NFI = 0.948, RMSEA = 0.064.

### 3.3. Reliability

Cronbach’s alphas for the total scale and for the severity and impairment subscales were 0.84, 0.78, and 0.78, respectively. The item-total correlations were all above 0.40, showing good internal consistency of the total scale as well as subscales (Table 4).

### 3.4. Convergent and Divergent Reliability

There was a strong correlation between the score obtained in the scale based on DSM-5 criteria and the SPS-R total score (r_s_ = 0.59, *p* < 0.001), SPS-R severity subscore (r_s_ = 0.47, *p* < 0.001), and SPS-R impairment subscore (r_s_ = 0.59, *p* < 0.001).

Divergent validity was demonstrated by relatively weaker correlations with the total DASS-21 total score (SPS-R total: r_s_ = 0.29, *p* < 0.001, SPS-R severity: r_s_ = 0.21, *p* < 0.01, SPS-R impairment: r_s_ = 0.33, *p* < 0.001), DASS-21 anxiety subscore (SPS-R total: r_s_ = 0.24, *p* < 0.01, SPS-R severity: r_s_ = 0.21, *p* < 0.01, SPS-R impairment: r_s_ = 0.22, *p* < 0.01), DASS-21 depression subscore (SPS-R total: r_s_ = 0.23, *p* < 0.01, SPS-R severity: r_s_ = 0.12, *p* = 0.14, SPS-R impairment: r_s_ = 0.31, *p* < 0.001), DASS-21 stress subscore (SPS-R total: r_s_ = 0.33, *p* < 0.001, SPS-R severity: r_s_ = 0.28, *p* < 0.001, SPS-R impairment: r_s_ = 0.36, *p* < 0.001), and OCI-R score (SPS-R total: r_s_ = 0.13, *p* = 0.11, SPS-R severity: r_s_ = 0.06, *p* = 0.44, SPS-R impairment: r_s_ = 0.20, *p* < 0.05).

### 3.5. Diagnostic Accuracy

To check the ability of the SPS-R scores to distinguish individuals with excoriation disorder from healthy individuals, the ROC analysis was utilized (see Figure 2) by assessing the area under the ROC curve (AUC). An AUC value > 0.9 (*p* < 0.001) was obtained, meaning that the SPS-R total score differentiates participants that declare meeting all of the DSM-5 criteria of excoriation disorder from healthy controls very well. Optimal cut-off point (SPS-R total score = 5) was determined using the highest Youden index calculated as follows: (Sensitivity + Specificity) − 1. Sensitivity, specificity, and accuracy measures for optimal cut-off point as well higher scores (for comparison) are presented in Table 5.

## 4. Discussion

The aim of the study was to examine psychometric properties of the Polish version of the Skin Picking Scale-Revised [11]. Specifically, we aimed to examine the factor structure, reliability, convergent and divergent validity, and diagnostic accuracy of the scale among individuals with self-reported skin picking disorder. The results of the study indicate that the Polish translation of the SPS-R has acceptable psychometric properties and can be used as a self-report tool for the measurement of both skin picking severity and picking-related impairment.

The results of the confirmatory factor analysis revealed that the two-factor structure of the SPS-R is consistent with the two-factor solution obtained for the original scale in an adult clinical sample [11] and for the German translation evaluated in adolescents [20]. This finding confirmed that the two separate SPS-R scores, reflecting skin picking severity and picking-related impairment, can be calculated on the basis of participants’ responses. It also indicated that the Polish version of the SPS-R can be applied in studies focusing on identifying factors that are related in a different way to the symptom severity and consequences (see [11]).

Our study also revealed a significant correlation between the SPS-R severity and impairment subscales. This finding is in line with the common-sense assumption that more severe picking results in more severe functional impairment. However, it also suggests that there may be one higher-order factor responsible for subscales association (see [11]). Unfortunately, in the current study, hierarchical CFA model could not be identified, as at least three first-order factors are needed for this purpose [21]. However, despite the lack of analyses concerning the presence of the second-order factor, high correlation (>0.7) between two first-order factors suggests that a total score of the SPS-R may be calculated in addition to the subscale scores. Likewise, the item-total correlation with the range from 0.452 to 0.678 showed high homogeneity of the items, thus a total score can also be used.

Our findings also indicate that the Polish version of the SPS-R revealed good internal consistency, and that the Cronbach’s alpha values obtained in the present study (0.83 for the total scale, 0.78 for the severity, and 0.77 impairment subscales) were very close to these reported for the original English version of SPS-R (0.83, 0.8, and 0.79, respectively; [11]), but lower than those yielded for the German translation (0.89, 0.85, and 0.87, respectively; [20]). The reason may be different characteristics of the samples. For the purpose of assessing the psychometric properties of the scale, we selected participants meeting similar criteria as in Snorrason et al.’s [11] study. On the other hand, in Gallinat et al.’s [20] research, the sample was not controlled for clinical characteristics—e.g., the participants with dermatological diseases were included in the study, and it was not assessed whether the skin picking behaviors declared by the participants were caused by those conditions. Choosing a more heterogeneous group could result in increased variance of items and higher Cronbach’s alpha coefficients in the German sample.

The convergent reliability was assessed by calculating the Spearman’s correlations between the three scores provided with the SPS-R and the score representing the DSM-5 skin picking (excoriation) disorder diagnostic criteria. Since the SPS-R items were extended to cover most of SPD symptoms, including functional impairment [11], we expected to observe a clear relationship between these two methods of assessing the severity of the disorder. Indeed, the findings revealed a strong, positive, and significant correlation between these two measurements confirming that individuals who obtained high SPS-R total score as well as high subscales scores, are also likely to meet the DSM-5 criteria for skin picking (excoriation) disorder.

Divergent reliability was examined by calculating Spearman’s correlations between the three SPS-R scores and DASS-21 which assess the current symptoms of depression, anxiety, and stress [13]. These analyses showed mostly weak associations between those constructs supporting the discriminant validity of the SPS-R. The findings are partly in line with Snorrason et al.’s [11] study which also revealed weak correlations between the DASS-21 subscales and the symptoms severity subscale of the SPS-R. Furthermore, similarly to Snorrason et al. [11], we observed moderate correlations between two of the DASS-21 subscales (depression and stress) and the impairment subscale of the SPS-R, suggesting higher frequency of psychopathological symptoms among individuals reporting that picking interferes with their daily life. However, the associations between the SPS-R impairment and the DASS-21 anxiety subscale obtained in the current study were weak, in contrast to Snorrason et al.’s study, where medium correlations between those subscales were detected. It is plausible that this discrepancy in results is due to the fact that most of the participants in our study declared a relatively low level of anxiety symptoms—50% of them obtained less than 6 out of 21 possible points on the anxiety subscale and none of them reached maximum score. In other words, differences in the effect sizes may be the product of lower variability and restriction of scores range in case of anxiety subscale.

The adequate divergent validity of the scale was also shown by the insignificant correlations between the SPS-R total score and severity score the OCI-R [14,15] measuring obsessive-compulsive symptoms frequency. Although obsessive-compulsive disorder and skin picking disorder may occur together, and studies suggest that they share some overlapping phenomenology, risk factors, and mechanisms [22,23,24], there are numerous differences between them. In contrast to OCD, skin picking is not preceded by the obsessional thoughts and can be performed without awareness. Usually, at the beginning, it is performed unconsciously and after a while individuals become more aware of it which differentiates skin picking behaviors from deliberate compulsive behaviors that are performed in order to reduce anxiety. Moreover, positive reinforcement plays a more important role in skin picking behaviors than in compulsions typical for OCD where negative reinforcement is more important [25]. The current study confirmed that the OCI-R and the SPS-R scales measure two distinctive phenomena and the score obtained in the one scale does not say much about the score in the other.

The ROC analysis showed that the total SPS-R score can distinguish individuals who declared that they meet all the DSM-5 diagnostic criteria for skin picking (excoriation) disorder from those who denied picking behaviors. In our study, a score of five or higher was accepted as a cutoff point for distinguishing SPD sufferers from healthy controls, with 100% sensitivity, 94% specificity, and 99% AUC. It is worth mentioning that in previous studies, higher SPS-R scores were used as indicators of pathological skin picking; for example, Gallinat et al. [20] adopted an SPS-R score of seven or higher as indicative of pathological skin picking, and Dixon and Snorrason [26] and Solley and Turner [27] used cutoff values of nine or higher. However, these cutoff scores were not established through ROC analysis. Although higher cutoff points may be recommended if it is important for us to avoid “false positives” (for example in scientific studies), a score of five or higher seems to provide the best balance between sensitivity and specificity and may be more useful in clinical practice (e.g., for initial screening).

The result of the study should be interpreted in light of some limitations. Firstly, our sample was a convenience sample consisting primarily of women. Although skin picking behaviors are more prevalent among women than men [5,17], and similar overrepresentation of women was observed in the previous validation studies [11,20], it may limit the generalizability of findings. Another important limitation is the sole reliance on self-report questionnaires: in future research, clinical assessment should be used when determining diagnostic accuracy of the Polish version of the SPS-R. Furthermore, the current study was cross-sectional—longitudinal studies are needed to assess the test-retest reliability of the scale and to check whether classifying participants on the basis of the SPS-R total score is reliable in the long term. Moreover, studies on clinical samples (with diagnoses confirmed by clinicians) are needed to facilitate research on the optimal cutoff score. Such studies could also determine the scale responsivity to change in symptoms observed during the course of disease and treatment. Because the research was carried out on an adult population, further studies are needed to investigate the psychometric properties of the scale among Polish adolescents.

## 5. Conclusions

In summary, in the study, we demonstrated robust psychometric properties of the Polish version of Skin Picking Scale-Revised and proposed optimal cutoff score that may indicate clinically relevant symptoms. Although some future research is needed, especially conducted in clinical samples, hopefully the adaptation of the scale will facilitate research on skin picking behaviors in Poland and understanding of this, still understudied phenomenon.

## Figures and Tables

**Figure 1 ijerph-19-02578-f001:**
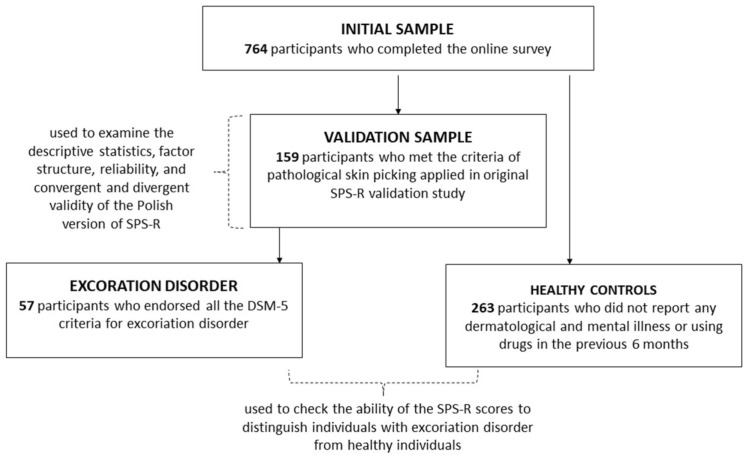
Selection of subgroups for the purpose of the statistical analyses.

**Figure 2 ijerph-19-02578-f002:**
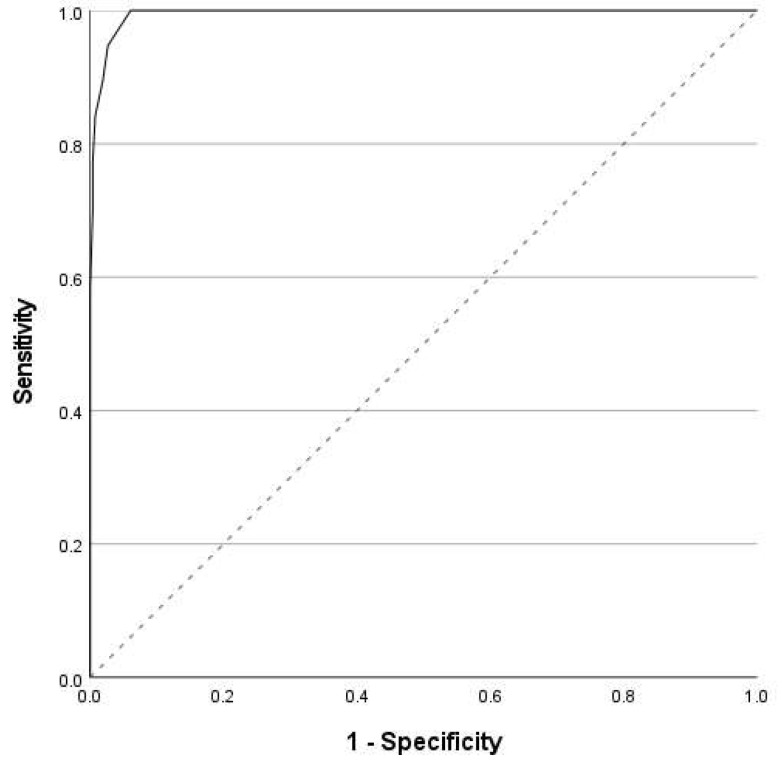
ROC curve for the SPS-R total score (individuals meeting DSM 5 excoriation disorder criteria vs. healthy individuals).

**Table 1 ijerph-19-02578-t001:** Sociodemographic characteristics of the subsample ^a^ (N = 159) used to determine psychometric properties of the SPS-R scale.

Variable		Women (N = 141)	Men (N = 15)	Total (N = 159)
Age	Mean (SD)	23.46 (5.78)	25.40 (10.22)	23.58 (6.29)
Marital status	N (%)			
Single		61 (43.26)	7 (46.67)	71 (44.65)
In relationship but not married		69 (48.94)	6 (40.00)	75 (47.17)
Married		11 (7.80)	2 (13.33)	13 (8.18)
Education	N (%)			
Primary school graduate		2 (1.41)	0 (0.00)	2 (1.26)
Secondary school graduate		3 (2.13)	0 (0.00)	3 (1.89)
Higher secondary school graduate		98 (69.50)	14 (93.33)	115 (72.33)
Vocational education		2 (1.41)	0 (0.00)	2 (1.26)
University graduate		35 (24.82)	1.00 (6.66)	36 (22.64)
PhD		1 (0.71)	0.00 (0.00)	1 (0.63)
Employment status ^b^	N (%)			
Student		110 (78.01)	5 (33.33)	115 (72.33)
Employed		63 (44.68)	12 (0.80)	75 (47.17)
Unemployed		3 (2.13)	0 (0.00)	3 (1.89)
Pensioner		4 (2.84)	0 (0.00)	4 (2.52)
Income (PLN)	N (%)			
<2000		96 (68.09)	11(73.33)	110 (69.18)
2000–3999		38 (26.95)	3 (20.00)	41 (25.79)
4000–5999		3 (2.13)	1 (6.67)	4 (2.52)
6000–9999		2 (1.41)	0 (0.00)	2 (1.26)
>10,000		2 (1.41)	0 (0.00)	2 (1.26)
Place of residence	N (%)			
Village/rural area		28 (19.86)	3 (20.00)	31 (19.50)
<20,000 inhabitants		14 (9.93)	3 (20.00)	17 (10.69)
20,000–50,000 inhabitants		12 (8.51)	1 (6.67)	13 (8.18)
50,000–100,000 inhabitants		14 (9.93)	2 (13.33)	16 (10.06)
100,000–200,000 inhabitants		9 (6.38)	0 (0.00)	10 (6.29)
200,000–500,000 inhabitants		12 (8.51)	1 (6.67)	14 (8.81)
>500,000 inhabitants		52 (36.88)	5 (33.33)	58 (36.48)

Note: ^a^ we selected participants meeting the criteria for pathological skin picking applied in the original validation study by Snorrason et al. [11]; ^b^ categories were not mutually exclusive.

**Table 2 ijerph-19-02578-t002:** Descriptive statistic.

	Mean (SD)						
Scale	Total Sample	Women/Men	Median	Skewness	Kurtosis	Observed Range	Possible Range
SPS-R	12.062 (4.553)	12.063 (4.526)/11.733 (5.035)	11.000	0.705	0.175	5–27	0–32
SPS-R severity	7.863 (2.687)	7.867 (2.625)/7.400 (2.898)	8.000	0.191	−0.394	2–15	0–16
SPS-R impairment	4.199 (2.487)	4.196 (2.501)/4.333 (2.664)	3.000	1.416	2.231	1–14	0–16
SPS-R item 1	2.037 (0.894)	2.030 (0.896)/1.930 (0.799)	2.000	0.405	−0.492	0–4	0–4
SPS-R item 2	2.280 (0.823)	2.280 (0.817)/ 2.200 (0.941)	2.000	−0.288	0.018	0–4	0–4
SPS-R item 3	1.584 (0.755)	1.590 (0.744)/1.400 (0.632)	1.000	0.948	0.197	0–4	0–4
SPS-R item 4	1.963 (0.968)	1.970 (0.911)/1.870 (1.356)	2.000	0.159	−0.509	0–4	0–4
SPS-R item 5	1.640 (0.926)	1.680 (0.908)/1.400 (1.121)	1.000	0.492	−0.358	0–4	0–4
SPS-R item 6	0.739 (0.848)	0.730 (0.857)/0.930 (0.799)	1.000	1.213	1.414	0–4	0–4
SPS-R item 7	0.453 (0.741)	0.430 (0.707)/0.670 (0.976)	0.000	1.654	2.205	0–3	0–4
SPS-R item 8	1.366 (0.677)	1.360 (0.687)/1.330 (0.617)	1.000	0.497	0.188	0–3	0–4
OCI-R	17.925 (11.589)	17.385 (11.523)/20.867 (10.690)	17.000	0.536	−0.512	0–47	0–72
DASS-21 total	27.379 (13.923)	27.322 (13.626)/25.867 (15.959)	28.000	0.163	−0.716	0–58	0–63
DASS-21 depression	9.366 (5.779)	9.315 (5.761)/8.800 (5.979)	9.000	0.189	−1.083	0–21	0–21
DASS-21 anxiety	6.988 (5.138)	7.028 (5.074)/6.200 (5.199)	6.000	0.616	−0.482	0–20	0–21
DASS-21 stress	11.025 (5.017)	10.979 (4.881)/10.867 (6.300)	11.000	−0.119	−0.759	0–21	0–21

Note: N = 159 (subsample selected on the basis of criteria applied in Snorrason et al. [11] study), Mann–Whitney test was used to compare whether there were differences in scores between women and men: all of the results were non-significant, SPS-R—Skin Picking Scale-Revised, OCI-R—Obsessive-Compulsive Inventory-Revised, DASS-21—Depression Anxiety Stress Scales 21-item version.

**Table 3 ijerph-19-02578-t003:** Factor loadings.

				95% Confidence Interval				
Factor	Item	Estimate	Std. Error	Lower	Upper	z	*p*	Standardized Estimate
Two-factor model								
SPS-R Severity								
	Item 1	0.678	0.060	0.561	0.795	11.352	<0.001	0.757
	Item 2	0.677	0.058	0.563	0.791	11.648	<0.001	0.818
	Item 3	0.472	0.054	0.366	0.578	8.735	<0.001	0.624
	Item 4	0.543	0.067	0.411	0.676	8.053	<0.001	0.562
SPS-R Impairment								
	Item 5	0.647	0.074	0.503	0.791	8.793	<0.001	0.696
	Item 6	0.640	0.085	0.472	0.807	7.493	<0.001	0.753
	Item 7	0.384	0.087	0.213	0.555	4.400	<0.001	0.516
	Item 8	0.489	0.051	0.389	0.590	9.512	<0.001	0.728
One-factor model								
SPS-R Total								
	Item 1	0.630	0.059	0.515	0.744	10.745	<0.001	0.703
	Item 2	0.639	0.055	0.531	0.748	11.588	<0.001	0.772
	Item 3	0.443	0.052	0.340	0.545	8.437	<0.001	0.585
	Item 4	0.524	0.065	0.396	0.652	8.024	<0.001	0.542
	Item 5	0.567	0.072	0.425	0.710	7.830	<0.001	0.611
	Item 6	0.546	0.081	0.386	0.705	6.715	<0.001	0.643
	Item 7	0.314	0.077	0.162	0.465	4.060	<0.001	0.422
	Item 8	0.418	0.050	0.320	0.515	8.409	<0.001	0.622

Note: N = 159; standardized estimate = factor loading.

**Table 4 ijerph-19-02578-t004:** Item reliability statistics.

		Cronbach’s Alpha		Cronbach’s Alpha
SPS-R	Item	If Item Dropped	Item—Total Correlation	Total Scale
Severity				0.783
	Item 1	0.680	0.679	
	Item 2	0.707	0.636	
	Item 3	0.722	0.613	
	Item 4	0.805	0.459	
Impairment				0.776
	Item 5	0.759	0.529	
	Item 6	0.683	0.650	
	Item 7	0.733	0.561	
	Item 8	0.713	0.614	
Total				0.836
	Item 1	0.809	0.624	
	Item 2	0.803	0.673	
	Item 3	0.821	0.530	
	Item 4	0.830	0.481	
	Item 5	0.819	0.557	
	Item 6	0.808	0.634	
	Item 7	0.830	0.452	
	Item 8	0.814	0.606	

**Table 5 ijerph-19-02578-t005:** Results of ROC analysis and optimal cutpoint.

								95% Confidence Interval		*p*
Cutpoint	Sensitivity (%)	Specificity (%)	PPV (%)	NPV (%)	Youden’s Index	AUC	SE	Lower	Upper	
**1**	**100**	**67.80**	**40.14**	**100**	**0.678**	0.995	0.002	0.990	1.000	<0.001
**2**	**100**	**74.62**	**45.97**	**100**	**0.746**					
**3**	**100**	**82.20**	**54.81**	**100**	**0.822**					
**4**	**100**	**87.88**	**64.04**	**100**	**0.878**					
**5**	**100**	**93.94**	**78.08**	**100**	**0.939**					
6	96.49	96.21	84.62	99.22	0.927					
7	94.74	97.35	88.52	98.85	0.921					
8	89.47	98.11	91.09	97.74	0.876					
10 ^c^	85.96	98.86	94.23	97.03	0.848					
11	84.21	99.24	96.00	96.68	0.835					
12	77.19	99.62	97.78	95.29	0.768					
13	75.44	99.62	97.73	94.95	0.751					
14	70.18	99.62	97.56	93.93	0.698					
15	57.89	100	100	91.67	0.579					

Note: Optimal cutpoint determined using Youden index was bolded; ^c^ none of the participants obtained 9 points on SPS-R.

## Data Availability

The data presented in this study as well as the Polish version of the SPS-R scale are available on request from the corresponding author.

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
