# Peer review of "Validation and Psychometric Properties of the Polish Version of the Skin Picking Scale-Revised"

_ijerph, 2022, doi:10.3390/ijerph19052578_

Round 1
Reviewer 1 Report
The manuscript shows potential to be published.
Regarding the "2.2. Measures" section,
a) Were the DASS-21 and OCI-R versions previously adapted to Polish population? If so, please cite the previous validation article.
b) Cronbach Alfa, CFI, NFI, NNFI, and RMSEA should be reported for all scales used with your sample, even if they were already adapted to the Polish population.
Regarding Appendix A,
Items should be published both in Polish and English
Author Response
We have revised our present research paper in light of Reviewer’s suggestions and comments:
Were the DASS-21 and OCI-R versions previously adapted to Polish population? If so, please cite the previous validation article.
In our study, we used the Polish translation of the DASS-21 by Makara-Studzińska et al., that is publicly available through DASS-21 homepage: http://www2.psy.unsw.edu.au/dass// . At the time of the study, and when the manuscript had ben written, the validation article describing psychometric properties of the Polish translation of the DASS-21 was not yet available, and therefore, we could not cite it in the manuscript. However, please note, that very recently, the work describing the Polish adaptation of the scale was published in the Frontiers of Psychiatry (Makara-Studzińska et al., 2022 ) and it shows good psychometrics properties of the Polish translation of the scale.
The OCI-R questionnaire was previously adapted to Polish population (Mojsa-Kaja et al., 2016). The validation article has already been cited in the submitted version of the manuscript (in the paragraph describing the OCI-R localized in the Method section) (see lines: 157, 167).
Cronbach Alfa, CFI, NFI, NNFI, and RMSEA should be reported for all scales used with your sample, even if they were already adapted to the Polish population.
As suggested, Cronbach’s Alphas, as well as fit statistics for the DASS-21 and the OCI-R obtained in the current sample were reported in the Method section of the manuscript (see lines 153-155 and lines 167-175).
Regarding Appendix A,
Items should be published both in Polish and English
Unfortunately, we do not have permission to publish original - English version of the scale. As Polish version may not be very informative for most of the readers we decided to delete the scale from the Appendix A. Instead, the scale will be available from the corresponding author on a request.
Reviewer 2 Report
I peer-reviewed the paper titled: Validation and psychometric properties of the Polish version of 2
the Skin Picking Scale-Revised
The authors aimed to validate the SPS-R for Polish patients.
The study is well written, and provides adequate methodology.
I have one major comment:
- As the sample size is analyzed and compared to one another in various tests I suggest to use a flow chart to visualize the initial cohort and subsequent selection of subgroups. Which were used as control for which comparisons? This could help the reader understand the studied participants a bit better.
Minor comments:
Line 199: JASP (version) and JAMOVI (version) - should be filled in.
I suggest (and this is more of an editorial suggestion) to use numeric representation when reporting number over 10 (i.e., 764 participants) and use written words to report only numbers under 10 like two or five (in the text, not tables).
Additionally, The results and methods are somewhat inconsistently reported. For example Analysis plan is reported in results section, when it is clearly methods, the same goes for sample size determination.
Other than that, the study is well designed and meticulously conducted.
Author Response
We have revised our present research paper in light of Reviewer’s suggestions and comments:
The study is well written, and provides adequate methodology.
Thank you for your very positive and encouraging feedback.
I have one major comment:
As the sample size is analyzed and compared to one another in various tests I suggest to use a flow chart to visualize the initial cohort and subsequent selection of subgroups. Which were used as control for which comparisons? This could help the reader understand the studied participants a bit better.
As suggested, in the revised version of the manuscript we added the flow chart (Figure 1) to visualise selection of groups to different analyses (see page 4).
Minor comments:
Line 199: JASP (version) and JAMOVI (version) - should be filled in.
As suggested, we added the information about the version of the software used for statistical analyses (see line 240).
I suggest (and this is more of an editorial suggestion) to use numeric representation when reporting number over 10 (i.e., 764 participants) and use written words to report only numbers under 10 like two or five (in the text, not tables).
We change the way the numbers were reported, as suggested.
Additionally, The results and methods are somewhat inconsistently reported. For example Analysis plan is reported in results section, when it is clearly methods, the same goes for sample size determination.
As suggested, paragraph describing analysis plan and sample size determination were moved to the Method section of the manuscript (see page 6).
Other than that, the study is well designed and meticulously conducted.
Thank you for your positive and encouraging feedback.